# A synthetic RNA editing factor edits its target site in chloroplasts and bacteria

Santana Royan[1], Bernard Gutmann[2], Catherine Colas des Francs-Small [2], Suvi Honkanen [2,3], Jason Schmidberger[1], Ashley Soet[1], Yueming Kelly Sun[2], Lilian Vincis Pereira Sanglard[2], Charles S. Bond[1] & Ian Small [2✉]

Members of the pentatricopeptide repeat (PPR) protein family act as specificity factors in C-to-U RNA editing. The expansion of the PPR superfamily in plants provides the sequence variation required for design of consensus-based RNA-binding proteins. We used this approach to design a synthetic RNA editing factor to target one of the sites in the *Arabidopsis* chloroplast transcriptome recognised by the natural editing factor CHLOROPLAST BIO-GENESIS 19 (CLB19). We show that our synthetic editing factor specifically recognises the target sequence in in vitro binding assays. The designed factor is equally specific for the target *rpoA* site when expressed in chloroplasts and in the bacterium E. coli. This study serves as a successful pilot into the design and application of programmable RNA editing factors based on plant PPR proteins.

[1] School of Molecular Sciences, The University of Western Australia, Crawley, WA, Australia. [2] Australian Research Council Centre of Excellence in Plant Energy Biology, School of Molecular Sciences, The University of Western Australia, Crawley, WA, Australia. [3] Synthetic Biology Future Science Platform, CSIRO, Canberra, ACT, Australia. ✉email: ian.small@uwa.edu.au

RNA editing is a term denoting a type of post-transcriptional modification of a transcribed RNA such that it differs from the sequence predicted from the genomic DNA[1]. This process occurs in diverse forms across all kingdoms of life, including the deamination of adenosines to inosines (which are read as guanosines in the mRNA) by ADAR enzymes in animals[2], the deamination of cytidines to uridines by APOBEC enzymes in animals[3] and by PPR-DYW proteins in plants[4], and the unknown mechanism of uridine modification to cytidine in an as yet unelucidated pathway present in some groups of plants (hornworts, lycophytes, and some ferns)[5,6]. Recently, several tools have been developed to perform single-nucleotide editing of mRNA[7]. These tools utilise the ADAR deaminase domain as the base modification tool, with specificity encoded in a programmable guide RNA[8,9]. The reliance on RNA-guided enzymes, whilst facilitating design, has some drawbacks (for example, it makes it hard to target organelles) and these artificial protein fusions have poor spatial accuracy around the desired target site due to flexibility in the linker region[8,9]. These limitations justify examining the possibility of redesigning natural protein-guided editing systems such as those found in plants.

RNA editing in plants is carried out by PPR proteins, a huge family of sequence-specific RNA-binding proteins with many roles in organellar RNA processing[10]. Of particularly relevance to RNA editing are the PLS-class PPR proteins, which generally contain triplets of three different types of PPR motifs in a (P1-L1-S1)$_n$-P2-L2-S2 arrangement[11]. These PPR arrays are followed by PPR-like E1 and E2 helix-turn-helix motifs, extending the characteristic repeating structure and a ~136 aa domain (named the DYW domain) with a conserved C-terminal D, Y, W tripeptide[11,12]. This DYW domain contains a cytidine deaminase-like active site and is necessary and sufficient for editing to occur[13–18]. PLS-class PPR protein-mediated editing has been characterised in several model plants, notably *Arabidopsis*, the moss *Physcomitrella patens*, rice, and maize[4] but is far more prevalent in some other little-studied plants, including hornworts, lycophytes, and ferns[5,6,19–23].

Correlation of PPR protein sequences with their target sites has led to a proposed 'code' for identifying interactions of PPR motifs with an RNA base[24–27]. These specificity-determining interactions involve hydrogen bonding between the amino acids at the 5th and last positions in each motif and the aligned RNA base[28]. With minor variations this code can be used to describe and predict interactions of any of the P, L or S motifs with RNA[26]. This code has been employed to make many successful predictions about which editing factors target which sites in mitochondria and chloroplasts[25,26,29,30]. It has also been used in attempts to modify target recognition by editing factors. CLB19 is an *A. thaliana* chloroplast editing factor required for editing of the *rpoA-78691* and *clpP1-69942* sites[31] that acts as the targeting factor in a complex with the DYW donor protein DYW2 and another editing cofactor, NUWA[32,33]. We previously showed that the target preference of CLB19 could be modified in vitro and in vivo[34] although we were unable to make it entirely specific for one or other of the two target sites.

The modular nature of PPR proteins, their large diversity, and the predictability of their interactions with RNA has prompted the design of synthetic PPR proteins based on consensus motifs, as first performed by[35]. Such synthetic proteins have proved to be useful tools for understanding the structure of PPR proteins and demonstrating their potential as programmable RNA-binding proteins[28,36–40]. Recently, the first crystal structure of a designer PLS protein[41] based on the consensus from genomic PLS sequences has revealed a similar structure to natural and synthetic P-class PPR proteins. This work suggests that the L-motifs within this synthetic protein are slightly misaligned with the P and S

motifs, but this misalignment is rescued by the formation of a complex with the cofactor MORF9 to allow for effective RNA recognition[41]. MORF9 is one of 10 related proteins in *Arabidopsis* that have been identified as important RNA-editing cofactors in seed plants[42,43], but until this crystal structure was published, it had been unclear how these proteins influence the process. MORF proteins are only one of several classes of cofactor theorised to form the 'editosome' in angiosperm organelles[44,45]. The apparent complexity of the editing process in angiosperms contrasts with its apparent simplicity in other plants where MORF proteins are absent and PPR editing factors are able to edit RNA on their own, as demonstrated by the reconstitution of editing in bacteria[18] and in vitro[17] with editing factors from *Physcomitrella*. These recent results are promising developments on the road to utilising PPR-RNA-editing factors as programmable biotechnological tools but to go further we need to be able to design new editing factors, which is facilitated by the use of synthetic proteins based on consensus motifs. We thus set out to construct a functional editing factor created from synthetic PPR motifs. As a test of functionality, we aimed to complement the well-studied clb19 mutant[31,34]. As a test of specificity, we aimed to target only one of the two editing events that are missing in *clb19*, thus attempting to design a synthetic editing factor that is more specific than the natural editing factor it is intended to replace. Here we demonstrate that a synthetic editing factor of this type can be expressed in plants and in bacteria, that it edits the chosen target RNA with high specificity, and that it is at least partially independent of plant-specific cofactors.

## Results

**Design and expression of a synthetic PPR-RNA-editing factor.** We designed a synthetic editing factor named dsn3PLS-DYW with the motif arrangement (P1-L1-S1)$_3$-P2-L2-S2-E1-E2-DYW (Fig. 1). This design was based on the most representative amino acids at each position in each motif based on 9730 PPR protein sequences from 38 different species, largely seed plant species, but also including *Physcomitrella patens*, *Selaginella moellendorffii*, and *Picea abies* (from data collated by[11]). The P1-L1-S1 triplets were designed to be position-specific, which means that the first P1 motif is slightly different to the second or the third P1 motif, and likewise for the L1 and S1 motifs (Supplementary Fig. S1). This was done in case the subtle position-specific differences found in natural editing factors are functionally important. The dsn3PLS-DYW protein was designed to start with a four amino acid cap composed of a starting methionine, followed by a glycine, asparagine and serine prior to the first P1 motif, based on work by ref.[46], who used the same cap in front of TPR motifs to stabilise α-helices. Compared to the only previously published design of a synthetic PLS-class PPR protein[41] (PLS)$_3$-PPR,[41] dsn3PLS-DYW has additional domains at the C-terminus, but lacks the extensive cap and solvating helix present in (PLS)$_3$-PPR that are derived from the N- and C-termini of *Zm*PPR10[28,47] (Supplementary Fig. S1). A key difference is also the residues at the 5th and last positions of each PPR motif, which in the case of dsn3PLS-DYW, were programmed to specifically target the *rpoA-78691* site recognised by CLB19, but not a second target of CLB19, *clpP1-69942* (Fig. 1).

The dsn3PLS-DYW protein was expressed as an N-terminal thioredoxin-hexahistidine-TEV protease site fusion in BL21 (DE3) *E. coli* and purified by nickel affinity purification. The N-terminal tag was cleaved off with TEV protease and the dsn3PLS-DYW further purified by reverse nickel affinity purification to about 85% purity (Supplementary Fig. S2). In parallel, a truncated version lacking the DYW domain (named dsn3PLS) and a short form of MORF9 (amino acids 75-196) were

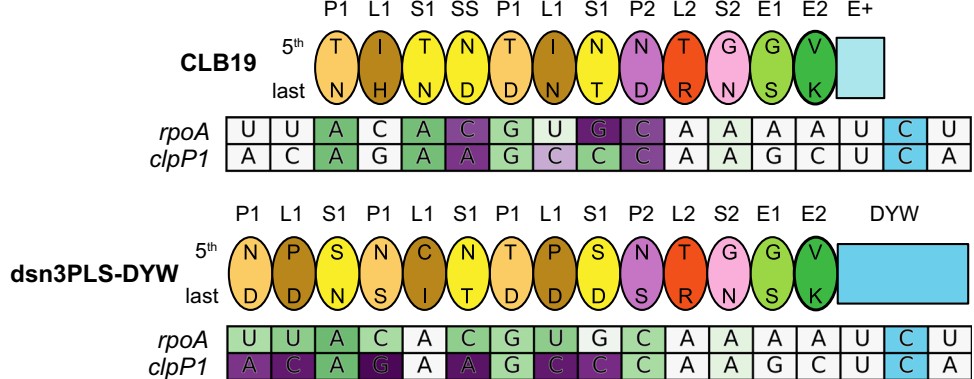

**Fig. 1 Schematic representation of CLB19 and dsnPLS-DYW bound to their target sites.** The proteins are represented by ovals for each PPR motif including the 5th and last specificity-determining amino acids. The target sites are coloured according to the predicted favourability of the alignment at each position, from favoured (green) to neutral (white) to disfavoured (purple). These values are taken from ref. [26]. The C at the editing site is shaded in blue.

purified similarly from *E. coli* Rosetta 2 (DE3) (Supplementary Fig. S2). RNA Electrophoretic Mobility Shift Assays (REMSAs) were performed on dsn3PLS and dsn3PLS-DYW with and without MORF9 protein, using Cy5-labelled oligonucleotide probes representing the *rpoA-78691* and *clpP1-69942*-binding sites of CLB19 (Fig. 2). In the absence of MORF9, dsn3PLS (and to a lesser extent, dsn3PLS-DYW) show slight binding of the *rpoA* target at the highest concentrations tested. The addition of MORF9 greatly enhances RNA binding, as previously observed for (PLS)₃-PPR[41] with optimal binding requiring at least 2 MORF molecules to each PPR molecule. The dsn3PLS/MORF9/RNA complex is clearly super-shifted with respect to the dsn3PLS/RNA complex. The formation of a dsn3PLS-DYW/MORF9/RNA complex was confirmed by analytical size-exclusion chromatography (Supplementary Fig. S3). No binding was observed to the *clpP1* target with or without MORF protein present, and MORF9 on its own displayed no binding to either target even at the highest concentrations.

**RNA editing in planta**. In order to test the RNA-editing capacity of the designer editing factor, dsn3PLS-DYW was introduced into *Arabidopsis thaliana* in a *clb19-3* mutant background. The construct used for plant transformation encodes a 60 amino acid rubisco small subunit (RbcS) transit peptide from pea (*Pisum sativum*) in order to facilitate chloroplast localisation of the protein. The *Arabidopsis CLB19* promoter and 5′ untranslated region (consisting of the 1 kb of DNA sequence from upstream of the start codon) was used to drive expression of the transgene. As a negative control, an almost identical construct was prepared, with the only difference being an inactivating mutation (E70A) in the DYW domain, which removes the conserved glutamate implicated as the catalytic residue in the deamination mechanism[14,15]. As a positive control, the native *CLB19* sequence (codons 34–500) was cloned into an equivalent construct. Plants homozygous for the *clb19-3* mutation are slow-growing and light-sensitive but were successfully transformed via *Agrobacterium*-mediated floral dipping. After hygromycin selection and growth, plants from the dsn3PLS-DYW lines displayed a greener phenotype than the *clb19-3* mutant line or the dsn3PLS-DYW (E70A) line, but they were less healthy than those complemented with *CLB19* (Fig. 3a). Hereafter, whenever we refer to 'plants expressing CLB19' or 'CLB19 plants', we are indicating transgenic *clb19-3* plants expressing this *RBCS-CLB19* construct that complements the *clb19* phenotype.

As a sensitive test of physiological state, RNA-seq data was obtained from leaves of plants grown under relatively low light

and analysed to quantify expression of chloroplast genes (Fig. 3b). In samples from plants expressing CLB19 or dsn3PLS-DYW, transcripts encoding components of the photosynthetic apparatus were generally more abundant than in *clb19* samples, whereas transcripts encoding components of the gene expression machinery showed the opposite pattern. Plants expressing dsn3PLS-DYW (E70A) showed an intermediate transcript abundance phenotype. Copious editing of both the *rpoA-78691* and *clpP1-69942* editing sites was observed in three biological replicates of the control line expressing CLB19 (Fig. 3c). Partial editing of *rpoA-78691* and no editing of *clpP1-69942* was observed in three biological replicates of the plant line expressing dsn3PLS-DYW. Thus, dsn3PLS-DYW is capable of editing *rpoA-78691* but not *clpP1-69942*. Very little editing of *rpoA-78691* was observed in the line expressing inactive dsn3PLS-DYW (E70A).

The transgene transcript abundance was quantified using the RNA-seq data to verify to what degree differences in editing could be explained by variations in expression level (Supplementary Fig. S4). The lines with the highest transgene transcript abundance also showed the highest proportions of edited transcripts, but the editing differences between the lines expressing CLB19 and dsn3PLS-DYW were much greater than the differences in transcript abundance. This suggests that CLB19 is more efficient at inducing editing, although it is possible that despite the lack of differences in transcript abundance, the proteins may accumulate to different levels. Despite the inclusion of a c-myc tag in the transgene constructs, we were unable to detect either the CLB19 or dsn3PLS-DYW transgene products by western blot to verify this.

**Off-target editing**. The extreme sensitivity of high-coverage RNA-seq allowed us to confidently detect off-target editing. The clearest example of this is a site in the second intron of *ycf3* (position 43350), which is a reported low-frequency site (12%) in wild-type Col-0[48], but which is edited at relatively high levels (~20%) in *clb19* plants expressing transgenic CLB19 (and not significantly edited at all in *clb19* plants without the transgene) (Fig. 4). The putative binding site for CLB19 upstream of this editing site closely matches expectations (Fig. 4b). This prompted us to search for other possible rare editing events that may represent 'off-target' events. All possible editing sites were screened for significant differences in editing between the transgenic lines and negative controls (*clb19* was used as a negative control for lines expressing dsn3PLS-DYW (E70A), and these in turn were used as negative controls for lines expressing CLB19 or dsn3PLS-DYW). Only one potential site, the intended target

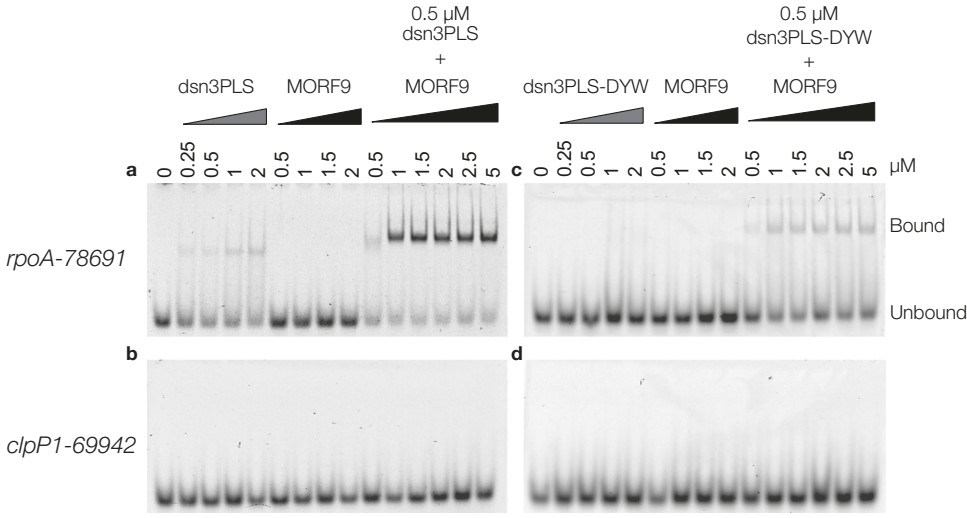

**Fig. 2 MORF-enhanced sequence-specific binding of Dsn3PLS proteins to *rpoA-789691* target RNA.** Binding assays were performed with dsn3PLS (**a**, **b**) or dsn3PLS-DYW (**c**, **d**) in comparison to, or together with, MORF9 in the presence of 1 nM *rpoA-78691* target RNA (**a**, **c**) or *clpP1-69942* (**b**, **d**).

*rpoA-78691*, was found in lines expressing dsn3PLS-DYW (E70A), with 24 events out of 2654 reads (0.89%). As the E70A mutation should completely prevent the deaminase activity of this protein, this residual activity is, at first sight, surprising. However, CLB19 completely lacks this part of the DYW domain and still effectively induces editing at this site by recruiting other cofactors, so it is possible that the dsn3PLS-DYW (E70A) can do the same, albeit with much reduced efficacy. Besides the *ycf3-43350* intron site already mentioned, twelve other sites were identified as potential off-target events induced by CLB19 (Fig. 4a) and eleven as potential off-target events induced by dsn3PLS-DYW, using an arbitrary threshold of 2 for the log(odds ratio). We believe that two of these sites are unlikely to be targeted by dsn3PLS-DYW or CLB19. The site 49209 is a known low-frequency editing event in the inter-cistronic region of *ndhK-ndhJ* transcripts, reportedly edited at ~6% in wild-type *Arabidopsis*[48]. Although we found this site to be edited at substantially higher levels in plants expressing CLB19 or dsn3PLS-DYW (~14%), it is also significantly edited in *clb19* (~2%) and dsn3PLS-DYW (E70A) plants (~2%), making it unlikely that it is a direct target of the introduced editing factors. The site 1502 is barely edited at all (0.03%) and at approximately equal levels in dsn3PLS-DYW, CLB19 and *clb19* plants. Excluding these two probable 'false positives', the other potential off-target sites show similarity to the *rpoA-78691* site (Fig. 4b). More details on these sites, including the amino acid changes they are predicted to cause, are given in Supplementary Table S1.

**RNA editing in bacteria.** To see if dsn3PLS-DYW was capable of editing RNA in a heterologous system, we expressed the protein in *E. coli* together with its target site, and optionally together with MORF2. Based on the work by Oldenkott et al.[18], the bacterial expression vector containing the dsn3PLS-DYW gene was modified to insert 39 bp regions covering either the *A. thaliana rpoA-78691* or *clpP1-69942* editing sites into the 3' untranslated region of the dsn3PLS-DYW transcript. Expression of this modified dsn3PLS-DYW transcript including a downstream editing site was induced together with MORF2 co-expressed from pETM11, or as a negative control, unmodified pETM11 (which expresses human max dimerisation protein 1). The presence of either pETM11-based plasmid greatly reduces dsn3PLS-DYW expression, presumably through sequestration of T7 RNA polymerase as the effect is primarily transcriptional. RNA editing was assessed by sequencing of cDNA derived from total RNA of 1 mL aliquots

of bacterial culture after 18 h growth post induction at 16 °C (Fig. 5). Under these conditions, editing of the *rpoA-78691* site was observed in the presence of MORF2 or MORF9, but only at reduced levels in the absence of MORF protein. This editing is abolished by the DYW mutation E70A, regardless of MORF co-expression. No off-target events were detected using the same significance thresholds as used in the analysis of chloroplast RNA. In order to identify if dsn3PLS-DYW was capable of uridine to cytidine 'reverse' editing, the *rpoA-78691* base was mutated at the DNA level in the plasmid to a thymidine, which would produce a uridine at this position in the transcribed RNA. No conversion of this site to a cytidine was observed in the sequenced cDNA for dsn3PLS-DYW co-expressed with MORF. No editing of the *clpP1-69942* site was seen under any conditions tested (Fig. 5C).

## Discussion

The synthetic editing factor dsn3PLS-DYW was designed to bind specifically to the region upstream of the *A. thaliana* chloroplast *rpoA-78691* editing site recognised by the natural editing factor CLB19. The REMSA results (Fig. 2), and the observed RNA editing in plants and bacteria expressing dsn3PLS-DYW (Figs. 3 and 5), show that this aim was achieved successfully. CLB19 and dsn3PLS-DYW differ quite considerably (only 45% sequence identity in the motifs aligned to the same nucleotides) and notably most of the residues known to be implicated in sequence recognition are different between these two proteins. CLB19 recognises one other major site in *Arabidopsis* chloroplasts, the *clpP1-69942* site originally reported in ref. [31]. This site is not detectably bound by dsn3PLS-DYW in vitro (Fig. 2) and not edited in plants or bacteria expressing dsn3PLS-DYW (Figs. 3 and 5). Hence by rational design we have succeeded in engineering an RNA-editing factor that is more specific than its natural counterpart, whereas our previous attempts to achieve the same goal by modification of the natural protein achieved a far less dramatic shift in specificity[34]. The fact that complementation of only the *rpoA* editing defect was sufficient to almost fully restore normal growth and chloroplast gene expression, at least under relatively low light conditions (Fig. 3), indicates that it is the loss of this editing event that primarily contributes to the strong phenotype of *clb19* mutants. Under higher light conditions growth defects become apparent even in plants expressing dsn3PLS-DYW (in a *clb19* background) (Fig. 3); we cannot be certain whether this is due to the lower degree of editing of the

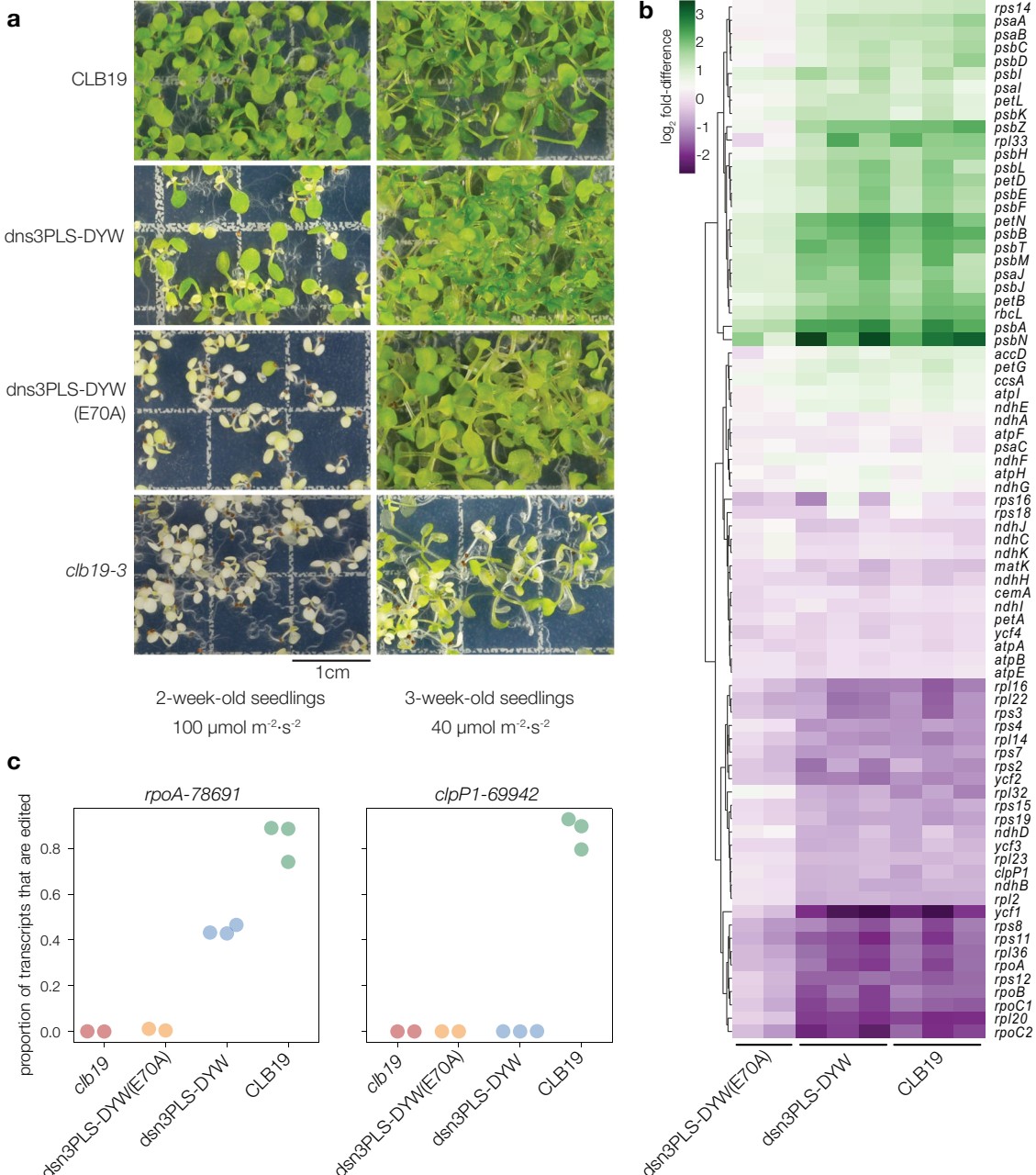

**Fig. 3 Visible and molecular phenotypes of *clb19* transformants. a** Seedlings were grown under a 16 h photoperiod with a light intensity of either 100 µmol m$^{-2}$ s$^{-1}$ (left panel) or 40 µmol m$^{-2}$ s$^{-1}$ (right panel). The construct used to transform the *clb19-3* mutant line is indicated on the left. The scale bar is 1 cm. **b** Chloroplast transcript abundances are shown as log$_2$ fold-difference compared to their levels in *clb19*. The RNA was extracted from plants initially grown under low light. **c** RNA editing at the *rpoA-78691* and *clpP1-69942* sites in the four genotypes.

*rpoA-78691* site than in wild-type plants or the complete lack of editing of the *clpP1-69942* site.

Interestingly, *clb19* plants expressing the inactivated dsn3PLS-DYW(E70A) construct were phenotypically distinguishable from untransformed *clb19* (Fig. 3), suggesting that they were partially complemented, and this was confirmed by the RNA-seq data indicating a low level (0.89%) of editing of the *rpoA-78691* site (Figs. 3 and 4). As this construct was not active in *E. coli* (Fig. 5), we presume this low level editing is due to a weak association between the dsn3PLS-DYW(E70A) protein and one of the DYW 'donor' proteins known to form complexes with other editing factors, such as DYW2 which forms a complex with CLB19[32,33]. From an evolutionary point of view, it is noteworthy that less

than 1% editing at a single site is sufficient to give rise to phenotypic differences that could provide a selective advantage. This might hint at how new editing events may arise and become selected for.

The high-coverage RNA-seq that detected editing of *rpoA-78691* in the dsn3PLS-DYW(E70A) samples also detected 'off-target' editing at numerous sites in plants expressing dsn3PLS-DYW and CLB19 (Fig. 4). Only one of these sites has been reported as an editing site previously[48], to our knowledge—the *ycf3-43350* site in intron 2 of the *ycf3* transcript. We do not think that this event has any functional significance as we did not detect any significant difference in *ycf3* intron 2 splicing (Supplementary Fig. S5) between plants expressing CLB19 (where *ycf3-43350* is

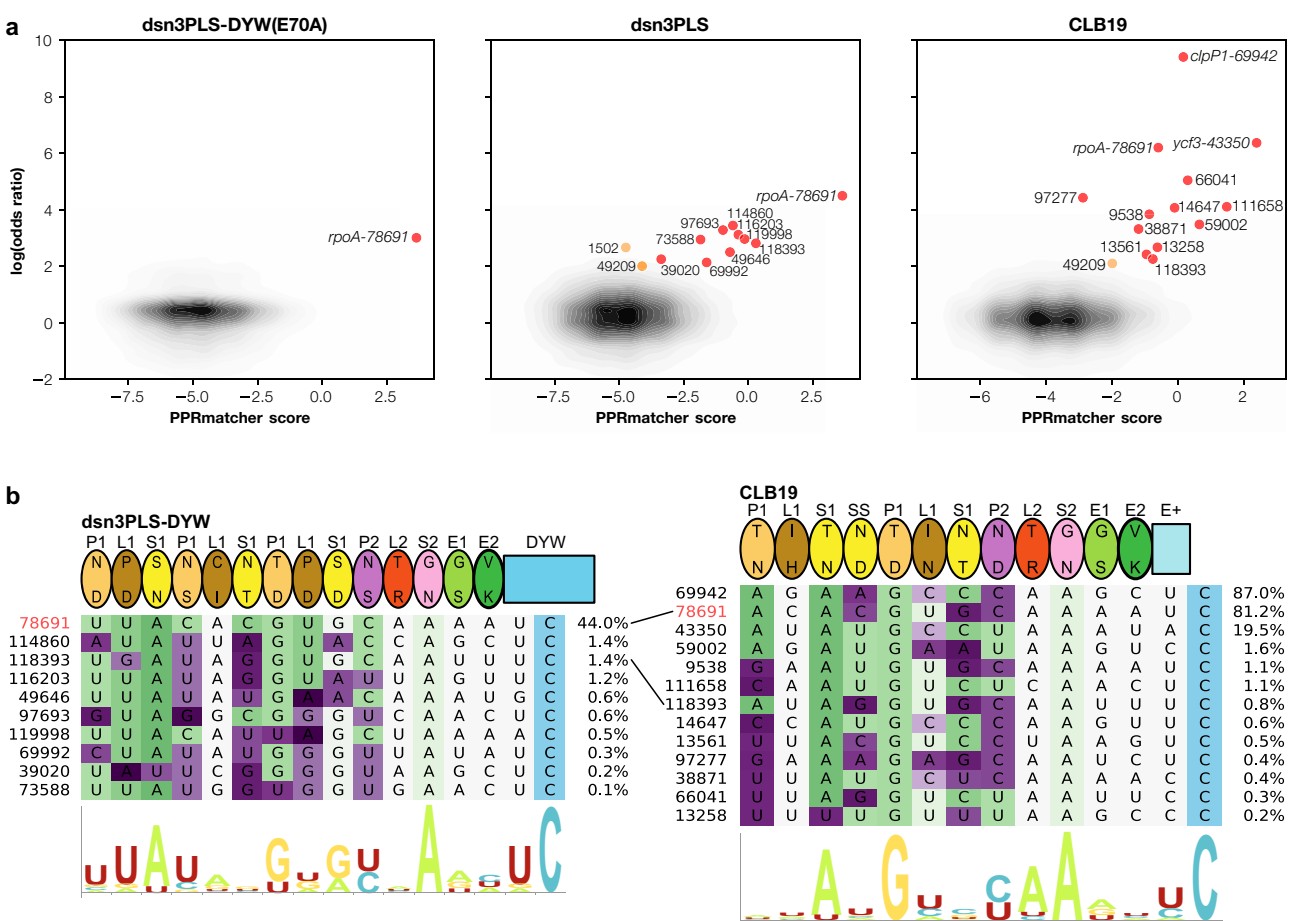

**Fig. 4 Putative off-target editing by dsn3PLS-DYW and CLB19. a** Potential editing sites plotted by their predicted binding to the expressed editing factor (x-axis) against the ratio of the editing activity compared to a negative control (expressed as the log of the odds ratio). Predicted binding scores were calculated with PPRmatcher (https://github.com/ian-small/PPRmatcher). For dsn3PLS(E70A) the negative control was *clb19*, for dsn3PLS-DYW and CLB19, the negative control was dsn3PLS(E70A). All sites where the difference in editing is statistically significant (Fisher exact test corrected for multiple testing) and the log(odds ratio) is greater than 2 are highlighted in red (or orange for two potential 'false positives' discussed in the text). The ~32,000 sites below this threshold are indicated by the density contours (grey-black). **b** Alignment of the highlighted sites from **a** with the corresponding editing factor. The *rpoA-78691* site is highlighted in red. The target sites are coloured according to the PPRmatcher score at each position, from favoured (green) to neutral (white) to disfavoured (purple). The percentage of edited transcripts at each site is indicated on the right. The sequence logos (constructed by Skylign[59]) indicate the nucleotide biases at each position.

edited) or dsn3PLS-DYW (where *ycf3-43350* is not edited). This site is specifically edited by CLB19, and according to our binding predictions (Fig. 4), is an even better match to CLB19 than either *rpoA-78691* or *clpP1-69942*, and yet is edited to a much lower extent. This may be due to a shorter half-life of the intron RNA, or poor accessibility within the highly structured intron, or because of the A at position −1 relative to the editing site; purines at this position are known to have an inhibitory effect on editing[18,49,50] and are rare in our collection of off-target events. Other off-target events were detected at much lower levels, below 2% (Fig. 4). Perhaps surprisingly, no off-target events were detected in bacteria, despite the much greater sequence complexity of the transcriptome and therefore the higher probability of close matches to the target site occurring by chance. We suspect that there may be two explanations for this; firstly, the read coverage of the chloroplast transcripts is generally much higher than for *E. coli* transcripts, allowing the detection of lower rates of editing; and secondly, whereas the *rpoA* target is a low abundance transcript in chloroplasts, the target transcript in the *E. coli* experiments is extremely abundant and may sequester a large fraction of the editing factor. Numerous off-target events were

observed when moss editing factors were expressed in *E. coli*[18], but there is a notable difference in the experimental conditions between their experiments and ours in that the moss editing factors do not require MORF proteins for activity. Co-expression of MORF2 or MORF9 in our experiments reduced PPR expression by more than an order of magnitude (simply due to competition between the expression plasmids for the expression machinery). Beyond this trivial explanation, it is however possible that there is a fundamental functional difference between the MORF-requiring proteins from angiosperms and the MORF-independent proteins from bryophytes, lycophytes and ferns. We have speculated, based on characteristic differences in L motif sequences[6], that the MORF-independent proteins have reduced specificity because their L-motifs are unable to distinguish between different bases. This is consistent with the moss editing factor results, where the off-target sites generally show any of A, C, G or U aligned with the L-motifs in their proteins[18].

We believe that the putative off-target events in chloroplasts are true editing events catalysed by the introduced editing factors because of the statistically significant difference in the transformed lines with respect to the controls, and because they are

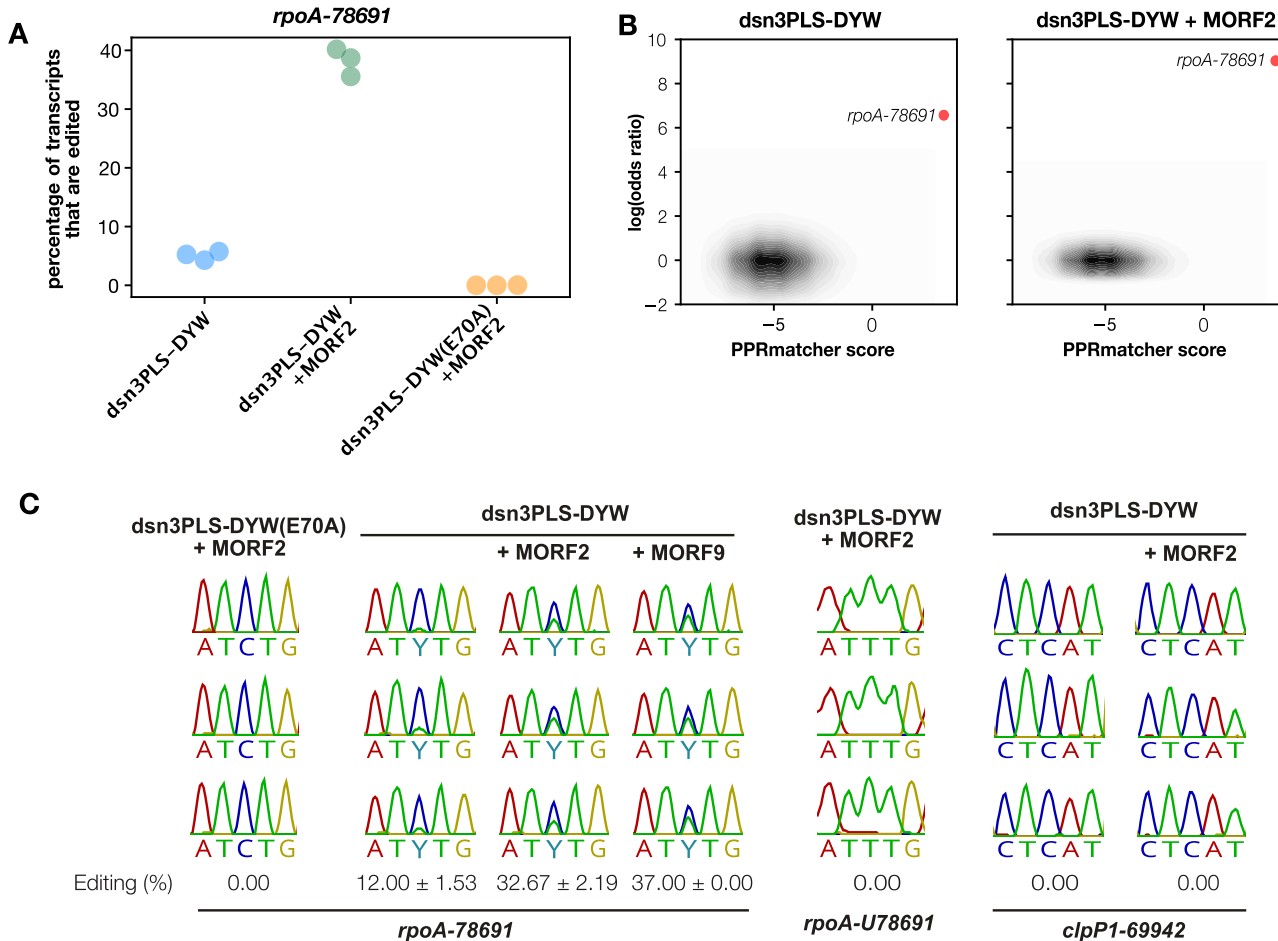

**Fig. 5 RNA editing in bacteria. A** RNA editing quantified by RNAseq at the intended target site in three independent samples of bacteria expressing dsn3PLS-DYW, dsn3PLS-DYW together with MORF2 or dsn3PLS-DYW(E70A), together with MORF2. **B** Potential editing sites plotted by their predicted binding to dsn3PLS-DYW (x-axis) against the ratio of editing activity compared to the negative control (dsn3PLS-DYW(E70A) together with MORF2), expressed as the log of the odds ratio. Predicted binding scores were calculated with PPRmatcher (https://github.com/ian-small/PPRmatcher). The only significant site where the log(odds ratio) exceeds 2 is the intended target site, *rpoA-78691*, highlighted in red. The ~492,000 sites below this threshold are indicated by the density contours (grey-black). **C** Additional tests of editing in *E. coli*. From left to right: dsn3PLS-DYW(E70A) + MORF2 as a negative control; MORF2 and MORF9 enhance editing by dsn3PLS-DYW by similar amounts; a U at the *rpoA-78691* editing position is not detectably reverse-edited to C; the *clpP1-69942* site is not detectably edited with or without MORF2. All experiments were repeated in triplicate, as shown.

generally consistent with expectations based on predicted binding by dsn3PLS-DYW and CLB19. The putative off-target sites for CLB19 are generally consistent with the in vitro analysis of the contribution of individual CLB19 motifs to target binding, notably the major contribution of the 2nd P1 motif (5/35 combination TD, recognising G)[34]. Promisingly for applications of synthetic editing factors, editing at off-target sites of dsn3PLS-DYW did not exceed 1.5% at any of the 9 sites we detected. Although eight of these events lead to non-synonymous changes in coding sequences (Supplementary Table S1), such low amounts of editing are unlikely to be consequential through any loss-of-function effect on the encoded protein. These off-target events are informative for the design of future synthetic editing factors as they provide information on the specificity of recognition (or the lack of it) of individual motifs. For example, the S2 motif aligns with an A in all 21 sites recognised by either dsn3PLS-DYW or CLB19 (Fig. 4), suggesting a hitherto unrecognised importance of this motif in determining site specificity. Other motifs proved to be less specific than expected; for example, the 5/35 combination NS, thought from previous data[26,27] to be relatively specific for C over U, did not prove to be when considering dsn3PLS-DYW off-target events, where in 12/18 cases NS motifs aligned with a U in

the target (Fig. 4). This type of data will be helpful for optimising future designs.

Binding of dsn3PLS-DYW to its target sequence in vitro and editing in vivo was strongly (but not completely) dependent on the presence of MORF proteins (Figs. 2 and 5). MORF2 and MORF9 were equally able to promote editing by dsn3PLS-DYW (Fig. 5), and the evidence suggests that the PPR-MORF-RNA complex that is formed contains multiple copies of the MORF protein (Fig. 2 and Fig. S3). Since the discovery of the association between MORF/RIP proteins and RNA-editing factors[51], the exact role of these cofactors in editing has been unclear. They have been variously proposed to associate with N-terminal PPR motifs[51] and/or C-terminal E motifs[52], and to exist as monomers and/or homo- or heterodimers[53]. It was long uncertain whether they acted by enhancing RNA binding, by enhancing formation of larger protein complexes (the 'editosome') or by enhancing the editing reaction (reviewed in ref. [4]). Our results are entirely consistent with those obtained with a different synthetic protein based on consensus P-L-S motifs[41]. Hopefully this confirmation that MORF proteins act by enhancing RNA binding by PPR editing factors helps remove some of the confusion concerning the role of MORF/RIP proteins in RNA editing.

The currently favoured biotechnological tools for RNA editing, the REPAIR and RESCUE systems comprised of a base editor coupled to a deactivated Cas13, achieve a high degree of specificity via the short guide RNA complementary to the target RNA, but once bound, show undesired promiscuity, potentially resulting in off-target editing of any deaminable bases within a window of at least four nucleotides on either side of the editing site[8,9]. In this study, we have demonstrated the potential of a synthetic editing factor that can edit with high specificity. Natural PPR editing factors are extremely precise—editing almost always occurs at the 4th nucleotide 3' of the nucleotide aligned with the S2 motif[25–27]. This is also the case for dsn3PLS-DYW, as all the off-target events observed are consistent with this positioning of the editing factor relative to the edited C; indeed, no editing was detected at adjacent C residues where this would have been possible (sites 49646 and 69992). Thus, it is reasonable to imagine that the specificity of 'designer' editing factors based on PPR-DYW scaffolds could ultimately exceed that of Cas13-ADAR fusions. On the other hand, designing the specificity of the PPR array remains complex due to the uncertain contributions of MORF cofactors and the C-terminal S2-E1-E2 motifs. Given that the *Physcomitrella* editing factors PPR56 and PPR65 can edit without MORF proteins present[17,18], it should be possible to design a synthetic editing factor that does not require cofactors for optimal specificity and editing activity. Of particular interest in this context are the monotypic S motif arrays found in putative editing factors in lycophytes[11], likely to be MORF-independent[6].

In conclusion, this work demonstrates the successful use of a synthetic PPR protein as an RNA-editing factor and lays the foundation for detailed structural and mechanistic studies into the mechanism of RNA editing. Designer PPR proteins represent an attractive multipurpose scaffold for targeted RNA binding, particularly as programmable RNA-editing factors.

## Methods

**Plant materials**. All plant genetics experiments performed in this study were undertaken on the *Arabidopsis thaliana clb19-3* T-DNA insertion line SALK_123752[31]. For general growth and propagation, seeds were germinated on 0.5 × MS plates (0.5 % sucrose) under dim light (40 $\mu$E m$^{-2}$ s$^{-1}$) in long day cycles (16 h light, 8 h dark at 25 °C with 45% humidity) for 3 weeks. When the leaves started greening, seedlings were transferred to 100 $\mu$E m$^{-2}$ s$^{-1}$ light. In order to perform transformation by flower dipping, pale green seedlings were transferred to soil until they were able to produce flowers. For the phenotypic analysis in Fig. 3a, the 'high-light' plants were germinated directly under 100 $\mu$E m$^{-2}$ s$^{-1}$ light to enhance the phenotypic differences between the lines. For the RNA-seq experiments, leaves were harvested during the period of growth at 40 $\mu$E m$^{-2}$ s$^{-1}$.

**Bacterial strains**. *Escherichia coli* strains used in this study were DH5α (Thermo Fisher Scientific, Waltham, MA, USA; https://www.thermofisher.com/) for cloning, BL21 (DE3) (Novagen, Merck KGaA, Darmstadt, Germany; https://www.merckmillipore.com/) for standard protein expression, and Rosetta 2 (DE3) (Novagen) for *E. coli* RNA-editing experiments.

**Plasmids**. Bacterial expression vectors pETM11 and pETM20[54] were a gift from Dr. Gunter Stier (EMBL, Heidelberg, Germany; https://www.embl.de). The plant expression vector used was pCAMBIA1390 (CAMBIA, Canberra, Australia; https://www.cambia.org).

**Oligonucleotides and primers**. Primers and unlabelled oligonucleotides (Supplementary Tables S2 and S3) were obtained from Integrated DNA Technologies (IDT; Singapore; https://www.idtdna.com) or Sigma-Aldrich (now part of Merck KGaA; Darmstadt, Germany), while labelled oligonucleotides were obtained from Integrated DNA Technologies (IDT, USA).

**Design and synthesis of designer editing factor sequences**. Domain sequences of putative DYW-subgroup editing factors from 38 land plant genomes containing three P1-L1-S1 motifs were obtained as previously described[11]. Motifs of canonical length for their type were selected. P1-L1-S1 triplets were categorised according to their position in their protein of origin (1st, 2nd or 3rd triplet) and each category aligned separately to construct position-specific consensuses. Consensus sequences of the aligned motifs were constructed with the EMBOSS cons tool[55] with plurality

0. The dsn3PLS-DYW protein was designed to start with a four amino acid Met, Gly, Asp, Ser cap, followed by three repeating modules of position-specific P$_1$-L$_1$-S$_1$ triplets, a P$_2$-L$_2$-S$_2$-E$_1$-E$_2$ module, and terminating in a consensus DYW domain (Supplementary Fig. S1). Amino acids residues at the fifth and last positions of the P1, L1, S1, and P2 motifs were selected to bind to the RNA sequence upstream of the *A. thaliana rpoA-78691* editing site, based on the 2 amino acid PPR code for those motifs[24–27], while the residues at the fifth and last positions of the L2, S2, E1 and E2 motifs were chosen to be identical to those of *A. thaliana* CLB19[31]. A synthetic gene encoding the dsn3PLS-DYW sequence was synthesised as a single gene block (GenScript, New Jersey, USA; https://www.genscript.com/).

**Cloning, bacterial expression and purification**. The dsn3PLS-DYW gene was cloned into the *NcoI* and *XhoI* sites of the bacterial expression vector pETM20[54] by Gibson assembly. Sequences encoding MORF2 (residues 73-193; C82S) and MORF9 (residues 75-196; C85S, C187S) were cloned into the *NcoI* and *XhoI* sites of the bacterial expression vector pETM11[54] by standard subcloning. The dsn3PLS-DYW expression vector was transformed into *E. coli* strain BL21 (DE3), while the MORF2 and MORF9 expression vectors were transformed into *E. coli* strain Rosetta 2 (DE3). Strains were grown in LB medium at 37 °C, 200 rpm to an absorbance of 0.6–0.8 at 600 nm, induced with addition of 0.1 mM IPTG, with addition of 20 $\mu$M ZnSO$_4$ to the dsn3PLS-DYW expressing cultures, and maintained at 16 °C, 200 rpm for 16–18 h. Bacterial pellets were resuspended in 40 mL of binding buffer (50 mM Tris-HCl pH 8.0, 150 mM NaCl, 1 mM DTT) supplemented with 125 units of Benzonase Nuclease (Sigma). Cells were lysed on ice using an Emulsiflex C5 high-pressure homogeniser (Avestin) to a maximum operational pressure of 16,000 psi until the lysate was clarified. The soluble fraction was separated from the insoluble fraction by centrifugation at 24,000 × g for 45 min at 4 °C and passed through a 0.22 $\mu$m syringe filter (Merck Millipore) before purification. Filtered bacterial lysate in binding buffer containing recombinantly expressed proteins was loaded onto a 5 mL His-Trap HP column (GE Healthcare) and the protein was eluted using a gradient of imidazole (20–500 mM) on an NGC Quest Plus (Bio Rad Laboratories) FPLC instrument (Supplementary Fig. S2). Fractions of interest were pooled and supplemented with 10 mM DTT and recombinant His-tagged TEV protease. The sample was transferred to a dialysis sac and dialysed against 50 mM Tris-HCl pH 8.0, 50 mM NaCl, 2 mM DTT at 4 °C for 16 h. The digest was then applied to the 5 mL HisTrap HP column (GE Healthcare) to remove the tagged protease and residual uncleaved protein (Supplementary Fig. S2). Purified MORF2 and MORF9 protein was then concentrated using a Vivaspin 3 kDa MWCO concentrator (GE Healthcare), while purified dsn3PLS-DYW protein was concentrated using a Vivaspin 10 kDa MWCO concentrator (GE Healthcare). Concentrated protein was divided into 200 $\mu$L aliquots, frozen in liquid N$_2$ and stored at −80 °C until further use.

**RNA electromobility shift assays**. REMSAs were performed as described previously[34]. Briefly, proteins purified were prepared on ice in a dilution series from 0.05–0.2 $\mu$M dsn3PLS-DYW or dsn3PLS-E2, or at a constant concentration of 0.1 $\mu$M dsn3PLS-DYW or dsn3PLS-E2, with a gradient of 0.1–0.5 $\mu$M MORF9 in 2 × REMSA binding buffer (85 mM Tris, 165 mM HEPES, 200 mM NaCl, 12.5 mM DTT, 0.25 mM EDTA, 1.25 mg mL$^{-1}$ heparin, 0.1 mg mL$^{-1}$ BSA). 1 nM of pre-heated diluted RNA oligonucleotide was added to the sample and incubated for 30 min at ambient temperature before loading onto a 5% polyacrylamide gel, running at 100 V. After 30 min, the fluorescence was visualised on an Amersham Typhoon system (GE Healthcare), with an excitation wavelength of 488 nm and an emission wavelength of 520 nm.

**Editing complementation in *A. thaliana***. The genes encoding dsn3PLS-DYW, dsn3PLS-DYW (E70A) or CLB19 (34–500) were cloned into pCAMBIA1390, along with the sequences for the native CLB19 promoter (−1000 to −1), the N-terminal chloroplast targeting peptide from pea RbcS, and a 4 × c-Myc epitope tag 5′ to the editing factor gene. Expression plasmids were transformed into 2-month-old *clb19-3* plants by floral dipping using *Agrobacterium tumefaciens* strain GV3101 Rif$^R$ Gent$^R$. Total RNA was isolated with RNAzol RT reagent (Sigma) from 3.5-week-old seedlings (T3) grown on half-MS plates under dim light conditions (40 $\mu$mol m$^{-2}$ s$^{-1}$).

**Plant RNA sequencing**. Two (for *clb19-3* and *clb19-3* + dsn3PLS-DYW(E70A) lines) or three (*clb19-3* + CLB19 and *clb19-3* + dsn3PLS-DYW lines) independent libraries were made from 250 ng of total RNA treated with Turbo DNase (Ambion, Thermo Fisher) using an Illumina TruSeq Stranded Total RNA with Ribo Zero (Plant) library preparation kit. The sequencing run (150 nt, paired ends) was performed on an Illumina HiSeq4000 sequencer by Novogene (Beijing, China; https://en.novogene.com). The read data is available from NCBI (BioProject ID PRJNA680434). Reads were first de-duplicated using *clumpify* (using parameters *dedup optical dist = 40*) from the *bbmap* package (https://sourceforge.net/projects/bbmap/), then trimmed of adapters with *bbduk* (parameters: *ktrim = r k = 23 mink = 11 hdist = 1 tpe tbo ftm = 5*) and mapped to the Col-0 plastid genome (accession AP000423) with *bbmap* (parameters: *ambiguous = random mappedonly = t*). To ensure that reads mapping across the arbitrary linear ends of the reference genome were not lost, the reference was extended by 1 kb prior to mapping. For analysis of chloroplast transcript abundances, mapped read pairs overlapping annotated CDS features were counted using a script based on

biojulia (https://biojulia.net) available at https://github.com/ian-small/pyrimid. Counts were normalised using a pseudo-reference approach taken from DESeq2[56]. For analysing editing, strand-specific nucleotide counts were obtained using the same in-house package (https://github.com/ian-small/pyrimid). PPR binding scores were calculated with PPRmatcher (https://github.com/ian-small/PPRmatcher) using the Kobayashi scoring table (derived from ref. [26]). For quantifying transgene transcripts, the de-duplicated, trimmed RNA-seq reads were mapped to Arabidopsis transcripts (TAIR10 annotations) supplemented with the relevant transgene sequence. The mapping and quantification were done with salmon 1.4.0[57], using the Arabidopsis nuclear, chloroplast and mitochondrial genomes as 'decoys'.

**Bacterial RNA-editing experiments**. *E. coli* RNA-editing systems were set up based on the methods described by Oldenkott et al.[18]. Sequences 33 bp upstream to 5 bp downstream of the *rpoA-78691* or *clpP1-69942* editing sites were introduced into the 3′UTR of dsn3PLS-DYW in the pETM20 expression plasmid by PCR. The plasmids were assembled using NEBuilder HiFi DNA Assembly Master Mix (NEB) according to the manufacturer's instructions. For editing assays, pETM20: dsn3PLS-DYW plasmids containing the editing sites were transformed into Rosetta 2 (DE3) cells, or co-transformed with the empty pETM11 plasmid, pETM11-MORF9 (75-196; C85S, C187S) or pETM11-MORF2 (73-193; C82S). Single colonies were used to inoculate 5 mL LB starter cultures with the appropriate antibiotics and grown at 37 °C, 200 rpm for 16 h. Two-hundred fifty microliters of the starter culture were then used to inoculate 50 mL LB expression cultures with the same antibiotics. The cultures were grown at 37 °C, 200 rpm to an $OD_{600}$ of 0.5–0.6. Cultures were then cooled on ice for 10 min prior to supplementation with 0.4 mM $ZnSO_4$, and induction with 0.4 mM IPTG. Cultures were incubated at 16 °C, 200 rpm for 18 h before harvesting. Total RNA was extracted from 1 mL of *E. coli* bacterial pellet using the Direct-Zol RNA extraction kit (Zymo Research; Irvine, CAL, USA) according to the manufacturer's instructions. Reverse transcription was performed on 2 μg of DNase-treated total RNA extract using Superscript™ III Reverse Transcriptase (Invitrogen) through random hexamer primers, according to the manufacturer's instructions. Reverse transcription polymerase chain reaction (RT-PCR) was performed with Q5 DNA Polymerase (NEB) according to the manufacturer's instructions, using the SynthSEQ.FOR and T7short primers. Sanger Sequencing of the RT-PCR products using the SynthSeq3.FOR primer was performed by Macrogen Inc. (Seoul, South Korea). Editing efficiency was determined from the raw sequencing chromatograms using the EditR webserver available at http://baseeditr.com/. For RNA-seq analyses, 500 ng of DNase-treated RNA was rRNA-depleted[58] and then used for preparing sequencing libraries with the Illumina TruSeq Stranded total RNA library preparation kit as recommended by the manufacturer. The libraries were sequenced on an Illumina HiSeq4000 sequencer by Novogene. In all, 29–34 million 150 nt paired-end reads were obtained for each sample. The read data is available from NCBI (BioProject ID PRJNA680433). Analysis of the data was carried out as described above for the chloroplast RNA data, except that the reads were mapped to the *E. coli* BL21 genome (accession CP010816) and the relevant pETM11 and pETM20 constructs.

**Statistics and reproducibility**. Nucleotide count data was analysed statistically with a Fisher exact test as implemented in the Python scipy.stats package and the *p*-values were corrected for multiple testing using statsmodels.stats.multitest.multipletests with the Simes-Hochberg procedure. Odds ratios were used to calculate and visualise differences in editing as they are an estimation of the ratio of editing activities in the two samples being compared. Odds ratios were calculated after adding a pseudocount of 0.5 to all observations to avoid division by zero. All experiments were replicated at least once (and in triplicate where shown). Replicates are biologically independent samples (independent transformants).

## Data availability
Data supporting the findings of this work are available within the paper and its supplementary information files. A reporting summary for this article is available as a supplementary information file. The datasets and materials generated and analysed during the current study are available from the corresponding author upon request. The sequencing data from this study is available from the National Center for Biotechnology Information Sequence Read Archive under the BioProject accessions PRJNA680433 and PRJNA680434. Source data for Fig. 2 and Supplementary Figs. S2, S4 and S5 are available from the Dryad data repository (https://doi.org/10.5061/dryad.b8gtht7c6).

## Code availability
Julia code used for counting potentially edited nucleotides in read alignments and for scoring PPR-RNA alignments is available from https://github.com/ian-small/pyrimid (https://doi.org/10.5281/zenodo.4628938) and https://github.com/ian-small/PPRmatcher (https://doi.org/10.5281/zenodo.4628933).

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

## Acknowledgements
This research was supported by the Australian Research Council (grants FL140100179 to I.D.S, DP150102692 and DP200102981 to C.S.B and I.D.S, CE140100008 to I.D.S, DE150101484 to B.G.) and the CSIRO Synthetic Biology Future Science Platform (fellowship to S.H.).

## Author contributions
I.S., C.B., S.R. and B.G. designed the study. S.R., B.G., C.C.d.F.S., S.H., J.S., A.S. and L.V.P.S. carried out the experimental work. Y.K.S. and I.S. analysed the RNA-Seq data. S.R. and I.S. wrote the manuscript with input from all co-authors.

## Competing interests
The authors declare no competing interests.
