## [Peer Review File · Communications Biology]

This manuscript has been previously reviewed at another journal that is not operating a transparent peer review scheme. This document only contains reviewer comments and rebuttal letters for versions considered at *Communications Biology*

REVIEWERS' COMMENTS:

Reviewer #1 (Remarks to the Author):

Comments for the Author:

I reviewed an earlier version of this manuscript for Nature Plants. As I wrote previously, the manuscript convincingly demonstrates that a synthetic PPR protein designed to mimic a natural RNA editing factor promotes RNA editing at the targeted site in vivo, and when coexpressed in *E. coli* together with a MORF protein cofactor. Additionally, in vitro data add to the evidence that MORF cofactors strongly enhance the RNA binding affinity of PPR editing factors. Analysis of off-target effects shows that the synthetic protein is remarkably specific: it is more specific in *Arabidopsis* than the natural protein it was designed to mimic, and it lacked detectable off-target editing in *E. coli*. The design of the protein entailed more nuance than had been used in prior synthetic PPR studies, the results are impressive, and the Discussion is clear, interesting, and touches on all of the relevant issues. The findings are significant and will be of broad interest to plant biologists, evolutionary biologists, and biotechnologists.

I had made several requests during the original review, and all of those were adequately addressed in this revised manuscript. However, I have additional suggestions to improve the clarity of the manuscript for those less familiar with this field, as summarized below.

(Note that I could not use line numbers to refer to locations in the manuscript because they did not display in my PDF.)

1. The text would benefit from proof-reading. Some examples:

p. 3- reference formatting problem on the fifth line from the top.

- need a close parentheses on line 6 of Results.

- many misplaced or missing commas

- p. 5 last line: "but the phenotype was less healthy than those complemented"... mismatched singular (phenotype) versus plural (those).

2. A brief introduction to PPR proteins would be helpful when they are first mentioned at the top of page 2. This should include the fact that specificity is determined by amino acids at the fifth and last position of each PPR motif- which is necessary to understand the explanation of the protein design on p. 3.

3. Figure 1. Please label which amino acid is which (ie 5th or last) in the figure.

4. P. 7. " which is a reported low frequency site (12%) in wild type Col-0 but which is edited at relatively high levels (~20%) in plants expressing CLB19". This is confusing because Col-0 does express CLB19. Reword to make clear that the latter are *clb19* mutant plants expressing a CLB19 transgene.

A related point: The nomenclature "CLB19" is used in several places to refer to *clb19* mutants complemented with a CLB19 transgene (e.g. Fig 4 and text describing it on pp 7-8). However "CLB19" could be misinterpreted to be simply the wild-type CLB19 allele. Please use nomenclature that unambiguously indicates both nuclear genotype and the transgene.

Fig 4a and 5b legend: is "rate" the word that is intended ? Perhaps "ratio"? Or "extent". I could not understand what was plotted and would appreciate a clarified explanation.

p.8 bottom. When introducing the E. coli experiments, it would be helpful to explain the general approach before describing the details of the construct that was used.

Reviewer #2 (Remarks to the Author):

In this revised manuscript, the authors have taken proper care of the points raised in my initial review. Overall, this is a nice study that highlights the potential of engineering synthetic PPR proteins for site-specific editing of organellar transcriptomes in plants.

Reviewer #3 (Remarks to the Author):

Royan et al have convincingly created a synthetic PLS-class RNA editing factor with editing specificity for the rpoA-78691 site in the Arabidopsis chloroplast transcriptome. Transgenic clb19 mutant plants that express this synthetic editing factor have partially restored photosynthetic activity, as anticipated due to a second unedited site in the ClpP1 transcript in clb19 plants.

Overall, this is a well written manuscript with a nice demonstration of a highly specific, albeit not completely effective, engineered synthetic RNA editing factor. I have several thoughts that need clarification.

- 1) In figure 2, rpoA super-shifted bands in the dsn3PLS + MORF9 lanes appear more intense than in dsn3PLS-DYW + MORF9 lanes. Should this be interpreted as better binding of the dsn3PLS protein to the rpoA site than the dsn3PLS-DYW protein? If so, would transgenic over-expression of this protein in the clb19 background improve editing efficiency of the rpoA site?
- 2) In figure 3b, RNA-seq data is described comparing CLB19 complemented clb19 mutants, versus the dsn3PLS and dsn3PLS-DYW lines. Why wasn't RNA-seq performed in wild-type plants? Is it known that the CLB19 transgenic plants fully restore clb19 mutant phenotypes?
- 3) Discussion on page 7 suggests that higher dsn3PLS-DYW transcript abundance leads to higher editing rpoA site efficiency, though still not to the extent observed in CLB19 complemented plants. A suggestion that protein levels may be limiting for dsn3PLS-DYW is given. It seems equally likely that the dsn3PLS-DYW engineered protein does not function as well as CLB19 - perhaps the DYW domain interferes (based on the super-shift experiments?), is unnecessary (similar to the CLB19 mechanism) or ectopic expression changes the interaction with MORF proteins or other factors
- 4) The ycf3 off-target site is claimed to be edited at 12% in Col-O genotype but at 20% in CLB19 transgenic plants. Is this difference significant? What is the original genotype background in the clb19 mutant line? RNA-seq in the wild-type background line would have clarified this. Or, are there unknown factors in design or expression of synthetic RNA editing factors that could change the editing efficiency of unrelated editing sites?
- 5) The E70A mutation still results in low level but functionally significant amounts of editing at the rpoA site. Is this additional evidence that over-expression of a synthetic editing factor may alter natural editing response?
- 6) Increased editing of 49209 site relative to wild-type and clb19 mutants apparently also occurs, again suggesting ectopic expression of editing factors may have additional effects.
- 7) Figure 4b does not list the 49209 site data but does list the 43350 site data.
- 8) A detailed discussion of the relative insignificance of low-level editing of off-target sites is provided, while the very low level of rpoA editing (0.89%) observed in the E70A mutant line is shown to have functional significance. I agree that most off-target editing would likely be loss of function or innocuous and of little consequence, but it reinforces the above indications that the design in this case is still not perfect, and other synthetic editing factors could have more (or less) off-target effects.

REVIEWERS' COMMENTS:

Reviewer #1 (Remarks to the Author):

Comments for the Author:

I reviewed an earlier version of this manuscript for Nature Plants. As I wrote previously, the manuscript convincingly demonstrates that a synthetic PPR protein designed to mimic a natural RNA editing factor promotes RNA editing at the targeted site in vivo, and when coexpressed in *E. coli* together with a MORF protein cofactor. Additionally, in vitro data add to the evidence that MORF cofactors strongly enhance the RNA binding affinity of PPR editing factors. Analysis of off-target effects shows that the synthetic protein is remarkably specific: it is more specific in *Arabidopsis* than the natural protein it was designed to mimic, and it lacked detectable off-target editing in *E. coli*. The design of the protein entailed more nuance than had been used in prior synthetic PPR studies, the results are impressive, and the Discussion is clear, interesting, and touches on all of the relevant issues. The findings are significant and will be of broad interest to plant biologists, evolutionary biologists, and biotechnologists.

I had made several requests during the original review, and all of those were adequately addressed in this revised manuscript. However, I have additional suggestions to improve the clarity of the manuscript for those less familiar with this field, as summarized below.

(Note that I could not use line numbers to refer to locations in the manuscript because they did not display in my PDF.)

1. The text would benefit from proof-reading. Some examples:
 - p. 3- reference formatting problem on the fifth line from the top.
 - need a close parentheses on line 6 of Results.
 - many misplaced or missing commas
 - p. 5 last line: “but the phenotype was less healthy than those complemented” ... mismatched singular (phenotype) versus plural (those).

Thank you for catching these issues, we've corrected the errors that were noted, re-checked the formatting of all the citations and made the use of commas, italics etc more consistent.

2. A brief introduction to PPR proteins would be helpful when they are first

mentioned at the top of page 2. This should include the fact that specificity is determined by amino acids at the fifth and last position of each PPR motif- which is necessary to understand the explanation of the protein design on p. 3.

We have added a little more introduction to the top of page 2 and explicitly added that it is the 5th and last positions in the PPR motif that determine base recognition specificity.

3. Figure 1. Please label which amino acid is which (ie 5th or last) in the figure.

We have altered the figure as suggested.

4. P. 7. “ which is a reported low frequency site (12%) in wild type Col-0 but which is edited at relatively high levels (~20%) in plants expressing CLB19”. This is confusing because Col-0 does express CLB19. Reword to make clear that the latter are clb19 mutant plants expressing a CLB19 transgene.

We have rephrased the sentence as suggested.

A related point: The nomenclature “CLB19” is used in several places to refer to clb19 mutants complemented with a CLB19 transgene (e.g. Fig 4 and text describing it on pp 7-8). However “CLB19” could be misinterpreted to be simply the wild-type CLB19 allele. Please use nomenclature that unambiguously indicates both nuclear genotype and the transgene.

We have carefully explained at the bottom of page 5 exactly what we mean by ‘plants expressing CLB19’. We think this should be clear enough for readers without cluttering up the text and figures with complex nomenclature.

Fig 4a and 5b legend: is “rate” the word that is intended ? Perhaps “ratio”? Or “extent”. I could not understand what was plotted and would appreciate a clarified explanation.

We have rephrased the figure legends and added the following sentence to the Methods: ‘Odds ratios were used to calculate and visualise differences in editing as they are an estimation of the ratio of editing activities in the two samples being compared.’

p.8 bottom. When introducing the E. coli experiments, it would be helpful to explain the general approach before describing the details of the construct that was used.

We have added an introductory sentence to this section.

Reviewer #2 (Remarks to the Author):

In this revised manuscript, the authors have taken proper care of the points raised in my initial review. Overall, this is a nice study that highlights the potential of engineering synthetic PPR proteins for site-specific editing of organellar transcriptomes in plants.

Reviewer #3 (Remarks to the Author):

Royan et al have convincingly created a synthetic PLS-class RNA editing factor with editing specificity for the rpoA-78691 site in the Arabidopsis chloroplast transcriptome. Transgenic clb19 mutant plants that express this synthetic editing factor have partially restored photosynthetic activity, as anticipated due to a second unedited site in the ClpP1 transcript in clb19 plants.

Overall, this is a well written manuscript with a nice demonstration of a highly specific, albeit not completely effective, engineered synthetic RNA editing factor. I have several thoughts that need clarification.

We thank the reviewer for their comments and questions. We agree with their conclusion that whilst highly specific (albeit not perfect), the efficacy of editing by such engineered factors can probably be improved in the future.

1) In figure 2, rpoA super-shifted bands in the dsn3PLS + MORF9 lanes appear more intense than in dsn3PLS-DYW + MORF9 lanes. Should this be interpreted as better binding of the dsn3PLS protein to the rpoA site than the dsn3PLS-DYW protein? If so, would transgenic over-expression of this protein in the clb19 background improve editing efficiency of the rpoA site?

Yes, dsn3PLS appears to bind more rpoA in vitro than dsn3PLS-DYW does. This is rather contrary to what is seen for natural PPR proteins where inclusion of the DYW domain if anything improves RNA binding (e.g. Okuda et al, Plant J, 2014). As the primary RNA-binding portions of both proteins are identical, this is likely to be due to a negative effect of the synthetic DYW domain, perhaps an increased tendency to aggregation. This is line with the reviewers point that the synthetic editing factor is not yet an optimal design. As we believe this is probably an uninteresting artefact and any discussion would necessarily be extremely speculative, we prefer not to expand on this in the manuscript. As we have not used the dsn3PLS construct in any of the other experiments, this result has no bearing on any of the subsequent conclusions. The effect of dsn3PLS on editing in vivo may well be negative as it would act as a competitor for the rpoA site and is incapable of editing alone. The only way it could increase editing would be if it is capable of recruiting a DYW donor protein (as we postulate is occurring when we express dsn3PLS-DYW (E70A)), but we have no evidence that it can do this.

2) In figure 3b, RNA-seq data is described comparing CLB19 complemented clb19 mutants, versus the dsn3PLS and dsn3PLS-DYW lines. Why wasn't RNA-seq performed in wild-type plants? Is it known that the CLB19 transgenic plants fully restore clb19 mutant phenotypes?

We felt that clb19 complemented with a CLB19 transgene construct identical to the one used to express dsn3PLS-DYW was a more appropriate control than simply using wild-type plants. As we show in the paper, the molecular editing phenotypes of the clb19 mutant are fully restored in these complemented lines (in fact more than restored, editing is higher than in published studies on wild-type).

3) Discussion on page 7 suggests that higher dsn3PLS-DYW transcript abundance leads to higher editing rpoA site efficiency, though still not to the extent observed in CLB19 complemented plants. A suggestion that protein levels may be limiting for dsn3PLS-DYW is given. It seems equally likely that the dsn3PLS-DYW engineered protein does not function as well as CLB19 -

perhaps the DYW domain interferes (based on the super-shift experiments?), is unnecessary (similar to the CLB19 mechanism) or ectopic expression changes the interaction with MORF proteins or other factors

We entirely agree with the reviewer that 'it seems ... likely that the dsn3PLS-DYW engineered protein does not function as well as CLB19'. Indeed, the paragraph to which he/she refers states this explicitly 'this suggests that CLB19 is more efficient at inducing editing'. We give an alternative explanation (that dsn3PLS-DYW does not accumulate to the same level as CLB19) but we have no evidence for or against this alternative hypothesis and is not our preferred explanation for the results.

4) The ycf3 off-target site is claimed to be edited at 12% in Col-0 genotype but at 20% in CLB19 transgenic plants. Is this difference significant? RNA-seq in the wild-type background line would have clarified this.

The difference is certainly significant, but we can't be sure of the causality because we did not test wild-type plants in parallel, so the difference may be due to growth conditions rather than the genotypes. We think this is a minor issue that can remain unresolved without affecting any of our conclusions.

What is the original genotype background in the clb19 mutant line?

It's a SALK line (as stated in Methods), so Col-0.

Or, are there unknown factors in design or expression of synthetic RNA editing factors that could change the editing efficiency of unrelated editing sites?

Certainly, the most obvious being the requirement of dsn3PLS-DYW for MORF cofactors and thus the potential for competition with other editing factors.

5) The E70A mutation still results in low level but functionally significant amounts of editing at the rpoA site. Is this additional evidence that over-expression of a synthetic editing factor may alter natural editing response?

We assume that the E70A construct induces editing of rpoA by recruiting host editing factors, so it is possible that this affects editing at other sites that require those same factors. We don't see any clear evidence of this, though; apart from the rpoA site, editing in the E70A lines looks like editing in clb19. Any slight differences are probably more easily explained by secondary effects of the differences in chloroplast gene expression on RNA levels and turnover (because of the functional RpoA that can be produced in the E70A lines) rather than on direct effects on the editing machinery. But that's speculation, we have no evidence either way.

6) Increased editing of 49209 site relative to wild-type and clb19 mutants apparently also occurs, again suggesting ectopic expression of editing factors may have additional effects.

As for the previous point, we are sceptical that this is a direct effect of the introduced editing factors on the editing machinery and suspect that this is more likely to differences in plastid gene expression in the various lines. Transcripts encoding ndh transcripts are expressed very differently in green and yellow Arabidopsis leaves. The difference between our values observed for this site in the transgenic lines expressing dsn3PLS-DYW or CLB19 and the wild-type values reported by Ruwe et al may also be simply due to differences in growth conditions or age of the leaves or the plants.

7) Figure 4b does not list the 49209 site data but does list the 43350 site data.

As discussed in the text (bottom of page 7) we believe, based on the evidence from the different lines, that 49209 (and 1502) are false positives of the statistics pipeline and not true targets of either dsn3PLS-DYW or CLB19. Therefore, we think including this site in Figure 4b would just cause confusion. On the other hand, 43350 is clearly a target of CLB19 and thus merits inclusion.

8) A detailed discussion of the relative insignificance of low-level editing of off-target sites is provided, while the very low level of rpoA editing (0.89%)

observed in the E70A mutant line is shown to have functional significance. I agree that most off-target editing would likely be loss of function or innocuous and of little consequence, but it reinforces the above indications that the design in this case is still not perfect, and other synthetic editing factors could have more (or less) off-target effects.

We agree, but think that the discussion already covers these points in depth and in sufficient detail.